# An Exploratory Study of Electronic Word-of-Mouth Focused on Casino Hotels in Las Vegas and Macao

Mengying Tang [1,2] and Hak-Seon Kim [3,4,*]

1   School of Business, Cangzhou Normal University, Cangzhou 061000, China; tangmengying@ks.ac.kr
2   Department of Global Business, Kyungsung University, Busan 48434, Korea
3   School of Hospitality & Tourism Management, Kyungsung University, Busan 48434, Korea
4   Wellness & Tourism Big Data Research Institute, Kyungsung University, Busan 48434, Korea
*   Correspondence: kims@ks.ac.kr

**Abstract:** In order to investigate the key attributes of casino hotel customer eWOM and their structural relationships, this study selects two casino hotels located in Las Vegas and Macao. Through big data analytics, online reviews of two casino hotels from Google Travel were utilized. The frequency and CONCOR analyses showed the top 50 high-frequency words for each hotel and divided them into groups. The results of the factor analysis and linear regression analysis show that four factors, namely "Physical Environment", "Entertainment", "Experience", and "Amenity", in Las Vegas have a significant impact on customer satisfaction, while two factors, namely "Value" and "Physical Environment", do in Macao. Through the results, the study points out the general characteristics affecting customer satisfaction of casino hotels, as well as the distinctions in influencing factors of their customer satisfaction in different source markets.

**Keywords:** casino hotel; electronic word-of-mouth (eWOM); online review; customer satisfaction; text mining; semantic network analysis





## 1. Introduction

As the gaming industry burgeons, the development of mega casino resorts providing top-notch all-inclusive luxury entertainment is underway with progress, and it is concentrated in Las Vegas and Macao [1]. Casino hotels are not only providing gaming facilities and luxurious accommodation, but also numbers of non-gaming leisure activities and entertainments (e.g., live show, concert, shopping and dining, spa, golf, exhibition, etc.), and growing to be contemporary integrated resorts [2]. Casino hotels are diversifying their enjoyment products from gaming to non-gaming pleasure services to attract more non-gaming-oriented tourists rather than gamblers [3]. From the perspective of consumers, since service is intangible and perishable, consumers seek additional information to reduce the uncertainty and complexity of purchasing decisions [4]. Electronic word-of-mouth (eWOM) results from a summary of the customer's experience and is usually written voluntarily, without any economic cost or external stimulus [5,6]. It has become a major source of information [7]. Therefore, hotel managers need to understand the content of customer eWOM and adjust their service in light of eWOM's characteristics, so as to improve customer satisfaction and attract more customers.

Both Las Vegas and Macao are world-famous for their pillar industries of gaming and are in the process of diversified economic development [8,9]. Numerous studies have taken Las Vegas and Macao as objects to conduct comparative studies from different perspectives, which mainly focus on the gaming industry, including the gaming market [8,10], casino hotel business model [11], hotel staff management [12,13] and the theme of casino hotel [14]. In terms of eWOM, these studies are aimed to compare the development characteristics and status quo of the gaming industry in the two places and put forward feasible suggestions.

Combined with the concept of customer eWOM, most studies mainly focus on the satisfaction of customer eWOM in a single region or the impact of eWOM [15–17]. However, there are few special studies on customer eWOM in casino hotels; and the comparative research on customer eWOM in different tourist regions is also limited.

This study attempts to study the customer eWOM of casino hotels by extracting semantic keywords and their relevance from customers' online reviews. Moreover, it employs factor analysis to find out the key factors of customer online review and explores the relationship among influencing factors of customer satisfaction by taking customer score as satisfaction index. Based on the content of customer online reviews, this study is aimed at figuring out the influencing factors of customer satisfaction, as well as the relationship between the factors. Taking two Venetian hotels in Las Vegas and Macao as examples, this study, with the help of a big-data technology program named SCTM 3.0, collected customers' online reviews on Google Travel from October 2017 to September 2021; RStudio software was used to preprocess the data, Ucinet 6.0 was used for semantic network analysis, and SPSS software was employed for factor analysis and linear regression analysis.

Taking Las Vegas and Macao as research objects, both of which have developed a casino-hotel industry, this study aimed to discover the typical characteristics of customer eWOM in the casino-hotel industry and put forward more universal suggestions. With the same brand of casino hotels as cases, operated by the same group in Las Vegas and Macao, this study can make a more precise comparison of customers' different focus on casino hotels in these two places and provide the basis for the decision-making of different market segments of casino hotels and the hotel industry.

## 2. Literature Review

### 2.1. Gaming Industry and Casino Hotels

The gaming industry is now the pillar industry of some places, and casinos can provide major economic benefits [18]. Nevada was the first state in the United States to have legalized gaming in 1931 and as the most populous city in Nevada, Las Vegas, with the gaming industry as its center, has huge tourism, shopping, and vacation industries [19]. Thus, it is known as "Gambling City" and has become one of the world's famous vacation places and the most famous destination for travelers and gamblers on earth [20,21].

With a gaming history stretching back for more than three centuries, Macao was renowned as "Las Vegas of the East", and it appears in the world as the largest casino resort in terms of revenue [11]. Macao's gross gaming revenue has even exceeded the Las Vegas Strip in 2006, and more than 90% of Macao tourism revenue comes from casino gambling, making it the most prominent gaming city globally [22–24] that profits from the global economic upturn and support from China's government [24].

In another hand, the development of gaming industry brings economic benefits as well as negative effects, such as be detrimental to its aura as a destination for cultural or heritage tourism. It suggests not only fun and glitz but also vulgarity, sleaze, and tackiness [25]. Therefore, casino hotels in Macao strive toward transforming the casinos into larger, more wholesome Las Vegas–style integrated resorts that are capable of attracting conventions and tourists who will shop, eat, and enjoy cultural activities apart from gambling for a greater length of time [26].

Casino hotels are developing into integrated resorts with unique functions and purposes. An integrated resort is defined as a "'multi-dimensional resort that includes a casino, convention/exhibition centers, hotels and shopping and entertainment facilities" [27]. These contemporary casino hotels are trying to set a new standard of luxury by introducing the "over the top" architecture design, the most deluxe accommodation and casinos in the world, Michelin-quality restaurants, and high-class venues for convention business [28]. In casino hotels, apart from gaming, some non-gaming entertainments, such as live shows and pop concerts, also offer visitors with fascinating and intensive emotional experiences, such as excitement, challenging, etc.

Non-gaming entertainment is becoming more and more important for generating revenue rather than merely boosting the gaming volume [29]. Entertainment, leisure, and recreational activities are sharing the same value of providing casino-hotel visitors with hedonic experience; thus, they are all part of casino hotels [30]. The hospitality industry increasingly offers theme-based experiences that follow the Disneyland model [14,31]. The Venetian Hotel, an ideal example, is a mega casino resort brand owned by the Las Vegas Sands Corporation. It uses one world-renowned tourist destination as a consistent theme by replicating its perceptible form on a one-to-one scale. In this way, it embraces a range of miscellaneous functions that no longer rely solely on gambling [14]. With the theme of Venetian Water Town, it elaborately recreates the main landmarks of Venice, Italy. Moreover, the hotel's shopping center possesses abundant replicas of the Venetian Canal and other architectural features, with Venetian-style murals surrounding the hotel lobby and casino. Moreover, one of The Venetian Hotel's attractions is The Gondola Ride, whose gondoliers entertain visitors with Italian songs and anecdotes. This hotel also includes exhibition venues, restaurants, casinos, swimming pools, and other attractions [32].

Currently, there are two hotels: The Venetian Las Vegas, whose construction cost was $1.5 billion, and was the world's most expensive major resort when it opened to the public in 1999; and The Venetian Macao opened in 2007 and cost $2.4 billion to build. The Venetian Macao is the largest hotel resort in Asia and the second largest hotel resort around the world; additionally, it is also the first open resort on reclaimed land on the Cotai Strip. This building complex represents a turning point in the development of Macao. More importantly, The Venetian Macao not only changes the urban ecology of Macao, but also promotes Macao's development and determines its future development direction [14,32].

Io attempted to evaluate casino-hotel visitors' hedonic experience by exploring and examining their positive emotions with respect to their preference of hedonic activities and satisfaction at casino hotels [2]. Wong and Fong have researched the roles of three casino service quality drivers—game service, service Environment, and service delivery-on customer satisfaction and loyalty intention in Macao, the world gaming capital [33]. The majority of casino hotel studies concentrate on customer satisfaction, yet the research object is confined to a single hotel or one urban city. This study conducted a comparative study of Las Vegas and Macao casino hotels. Apart from exploring the general characteristics of the casino hotel's eWOM, it also compares customer features in different tourist source areas and influencing factors in customer satisfaction.

## 2.2. Electronic Word of Mouth (eWOM) and Online Review

Westbrook [34] propose that word-of-mouth (WOM) is the communication among consumers about products, services, or companies whose sources are considered independent of commercial influence. WOM communication is a process that consumers imitate and talk to each other according to the paradigm of social or alternative learning [35], which allows them to share information and opinions and guides purchasers to become interested in or give up specific products, brands, and services [36]. With the development of electronic informatization, Litvin et al. assume customer's electronic word-of-mouth (eWOM) can be defined as all informal communication aimed at consumers through Internet-based technologies and that these communications are in connection with the usage, characteristics, or sellers of particular goods and services [4]. Represented as evaluations of products on review websites, eWOM is considered to exert an impact on sales [37,38], and it includes likes, comments, ratings, comments, video recommendations, tweets, pictures, and blog posts [39]. By adding new online sources of information to the websites of existing hotels and travel companies, consumers are becoming key players in influencing others through online reviews [40]. Rosario et al. put forward that there are three eWOM stages, namely creation, exposure, and evaluation [41]. Positive eWOM can be amplified through owned and paid social media channels [42]. Thus, companies need to pay close attention to each dimension of eWOM so as to boost their business.

Multitudes of scholars have studied eWOM. Cantallops and Salvihave made an extensive review of the eWOM literature and divided it into two research routes [43]: the cause and effect of eWOM [44,45] and the influencing factors of eWOM [46]. Donthu et al., with bibliometric analysis and systematic review, pointed out that there are currently four main research topics in the eWOM area: determinants of eWOM, eWOM in hotel industry, cognitive aspects of eWOM, and service failure and recovery [42]. Nowadays, researchers are using the text-mining method to analyze eWOM. For instance, Ban and Kim used semantic network analysis to analyze the hotel package [47]. Nevertheless, a majority of previous studies are focused on the impact of the text review itself and applied text-mining techniques, aiming to extract meaningful knowledge from a variety of textual data and finding relationships and patterns within such unstructured information [48]. Studies on online reviews in the tourism and hotel industry mainly analyze how information exchange directly affects consumers' choice of a certain hotel [49]; most of them conclude that exposure to positive online reviews will increase the average probability of a consumer booking a room in the same hotel.

Online reviews have an effect on hotel performance [50]. A large number of online customer reviews greatly influence consumer purchasing decisions [51]. At present, in the hotel online review study, more attention is paid to the relationship between online review and online purchase, as well as that between satisfaction and online management; meanwhile, there is also research on the opinion mining of online reviews, the motivation for posting reviews, and the role of reviews [52]. Moro et al., using 504 reviews published in 2015, provided 21 hotels on the Las Vegas strip with a data-mining method for modeling TripAdvisor ratings. They prepared 19 quantitative features characterizing the reviews, hotels, and users and fed them into a support vector machine to model the score [53]. The recent study by Kim and Park, comparing both TripAdvisor scores and traditional customer satisfaction through travel intermediaries, found out that online reviews play a more significant role in explaining hotel performance metrics than traditional feedback [54]. This study aimed to analyze customers' main focus and influencing factors of the customer satisfaction of casino hotels through the online comments and ratings on Google Travel; meanwhile, it conducted the analysis from the perspective of the characteristics of Chinese and American tourists, who have different cultural backgrounds.

*2.3. Customer Satisfaction*

With the change of social environment, customer satisfaction has developed abundant theories. As for the research on customer satisfaction, it can be traced back to Cardozo's investigation on influencing factors of product satisfaction in 1965, which pointed out that customer satisfaction can directly affect their repeated purchase behavior [55]; thus, it is important for marketing to understand the factors influencing customer satisfaction. In order to adapt to the needs of various industries, it has given rise to the American Customer Satisfaction Index (ACSI), which is a type of market-based performance measure for firms, industries, economic sectors, and national economies [56]. Since online shopping is different from traditional shopping in many ways, the electronic Customer Satisfaction Index (e-CSI), a new index for measuring online customer satisfaction, has arisen [57].

Research on customer satisfaction can be classified into macro and micro parts [58]. On the one hand, macro studies show the relationship between satisfaction and the environment and reveal that different factors will have positive or negative effects on satisfaction. To illustrate this, Joung et al. investigated home-delivered-meals program in the relationship among service quality, satisfaction, and behavioral intention [59]; Lee et al. studied the effect of healthy food knowledge from the perspective of healthy foods' value, degree of satisfaction, and behavioral intention [60]; Hu et al., from the view of customers' expectations of hotels, studied the difference in the impact of attributes on satisfaction and re-sponsorship [61]. On the other hand, micro studies rely on customer feedback and communication, and they investigate the satisfaction model composed of influencing factors of satisfaction. In the service industry, they are reflected in studies on external customer satisfaction [62–64], and on

internal employee satisfaction in enterprises [65–67]. Consumers' satisfaction with a product or service is the result of a subjective comparison between their expectation and perception [68]. Taking online customer online reviews as independent variables and customer scores as their satisfaction index, this study attempted to explore the factors influencing customer satisfaction in regard to casino hotels in online reviews.

*2.4. Text Mining and Semantic Network Analysis*

Text mining refers to using information retrieval, information extraction, and natural language processing techniques to discover unknown useful patterns and knowledge in text [69,70]. Generally, the text-mining process consists of data collection, data extraction, data analysis, and other steps, and it includes the management information system [71]. The first step is to determine the information type the researcher is trying to find. Subsequently, it is crucial to limit the range of data to be collected and become familiar with the characteristics of keywords. Data extraction is the process of converting unstructured text data into the structured form. The analysis part is to analyze the text based on information extraction, clustering, and classification techniques, which are utilized as a management information system and accumulated as knowledge [72,73].

Text clustering is considered one of the most important topics in modern data mining. Text mining has been applied in many research fields. A large amount of the published research based on customer feedback and, particularly, in tourism and hospitality focus on the analysis of the textual contents from users' reviews through text mining and sentiment analysis. For instance, Cao et al. investigated the impact of online-review features hidden in the textual content of the reviews on the number of helpful votes of such review texts by applying text mining for extracting the review's characteristics [74], while Guo et al. applied text mining and topic modeling for unveiling several dimensions that hoteliers need to control for managing interactions with visitors [75]. However, several issues and challenges are brought up when it comes to using text mining. The most widely discussed are context specificities associated with the user and problem being dealt with, language barriers, and human communication issues, such as sarcasm and irony [76].

Semantic network analysis can be described as the use of network analytics techniques based on shared meaning as opposed to paired associations of behavioral or perceived communication links [77,78]. The text is encoded into a network to determine the structural relationships between words. Semantic network analysis can identify unrevealed intrinsic meanings in context [70,78]. In addition, word frequency and cluster analysis can assist in understanding the influencing degree of a particular word and analyzing how a word affects the relationship between groups [78–80]. As a quantitative text analysis method, semantic network analysis provides an impressive theoretical and methodological foundation to describe the semantic nature of online reviews [80]. Kang et al. examined current vaccine sentiment on social media by constructing and analyzing semantic networks of vaccine information from highly shared websites of Twitter users in the United States, which assisted public health communication of vaccines [81]. Ban and Kim conducted a semantic network analysis by collecting keywords of hotel packages through Google [47]. In 2021, based on the Korean Citation Index (KCI), Wang et al. made a series of studies to explore human resources management (HRM) trends by utilizing topic modeling and semantic network analysis [82].

## 3. Methodology

Customers' online reviews are an important source of eWOM. The Venetian Hotel has a significant advantage in the number of online reviews on Google Travel in Las Vegas and Macao. Based on customer online reviews of two case hotels, this study collected their customers' online reviews on Google Travel and explored their potential meanings through the results of various text mining. Semantic network analysis is based on the frequency of main words in online reviews, the relationships between main words, and the network structure. It can be used as a practical methodology for understanding the flow of web materials [83].

In order to find the characteristics of customer online reviews in casino hotels in Las Vegas and Macao, the process, performed in the theme modeling, mainly consists of three steps, shown in Figure 1. The first priority is to collect data that will be applied to topic modeling, that is, to use SCTM 3.0, which is the big-data technology program to collect online reviews and ratings for The Venetian in Las Vegas and Macao respectively. The period of collected data is four years, from October 2017 to September 2021. The second step is to carry out preprocessing and morphology analysis. The researcher processes the collected data with RStuido program and then transforms the unstructured data into a suitable format for theme modeling and visualizes the word frequency by Unciet 6.0. The last step, data analysis, is to perform a frequency analysis on the obtained words. In this study, the evaluation scores represent customer satisfaction, and SPSS software is used for factor analysis and linear regression analysis.

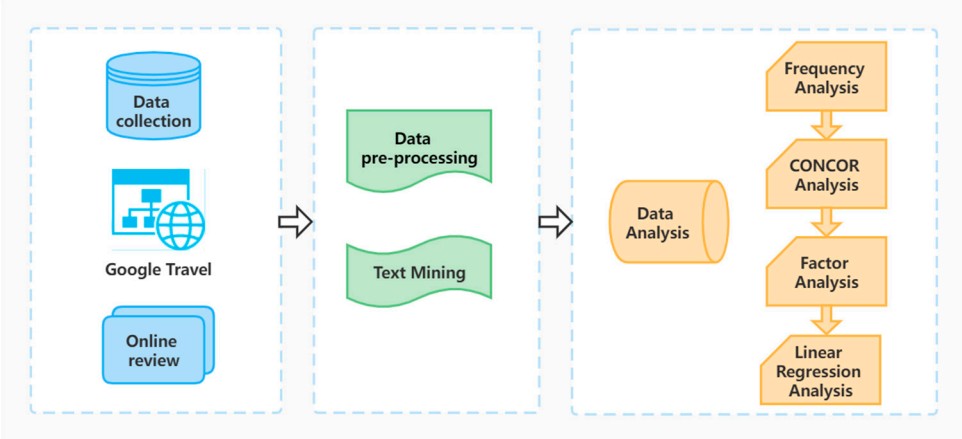

**Figure 1.** Research process.

## 4. Results

### 4.1. Frequency Analysis

Through Google Travel, this study collected 8591 and 8120 network reviews of The Venetian Hotel in Las Vegas and The Venetian Hotel in Macao, respectively. After deleting redundant, duplicate, or unnecessary words, we extracted fifty words from each data group. Table 1 summarizes the top 50 most frequently found keywords (100 words altogether) in online reviews for each hotel, and the frequent keywords are shown in Figure 2 with visibility.

**Table 1.** Top keywords' frequency of the case hotels.

| The Venetian Hotel, Macao | | | | The Venetian Hotel, Las Vegas | | | |
|---|---|---|---|---|---|---|---|
| Rank | Word | Freq. | % | Rank | Word | Freq. | % |
| 1 | Casino | 2258 | 9.37 | 1 | Room | 4454 | 11.47 |
| 2 | Good | 1745 | 7.24 | 2 | Great | 2913 | 7.50 |
| 3 | Room | 1637 | 6.79 | 3 | Beautiful | 2395 | 6.17 |
| 4 | Shopping | 1548 | 6.43 | 4 | Nice | 1797 | 4.63 |
| 5 | Beautiful | 1054 | 4.37 | 5 | Casino | 1775 | 4.57 |
| 6 | Big | 930 | 3.86 | 6 | Staff | 1438 | 3.70 |
| 7 | Luxury | 900 | 3.74 | 7 | Amazing | 1418 | 3.65 |
| 8 | Nice | 820 | 3.40 | 8 | Service | 1284 | 3.31 |
| 9 | Food | 808 | 3.35 | 9 | Clean | 1239 | 3.19 |
| 10 | Great | 792 | 3.29 | 10 | Restaurants | 1382 | 3.56 |
| 11 | Mall | 749 | 3.11 | 11 | Good | 1100 | 2.83 |
| 12 | Shop | 595 | 2.47 | 12 | Food | 1056 | 2.72 |
| 13 | Large | 560 | 2.32 | 13 | Shop | 1053 | 2.71 |
| 14 | Gondola | 531 | 2.20 | 14 | Suite | 942 | 2.43 |
| 15 | Amazing | 457 | 1.90 | 15 | Best | 877 | 2.26 |

**Table 1.** *Cont.*

| | The Venetian Hotel, Macao | | | | The Venetian Hotel, Las Vegas | | |
|---|---|---|---|---|---|---|---|
| 16 | Best | 447 | 1.86 | 16 | Pool | 835 | 2.15 |
| 17 | Huge | 447 | 1.86 | 17 | Shopping | 712 | 1.83 |
| 18 | Restaurant | 421 | 1.75 | 18 | Gondola | 657 | 1.69 |
| 19 | Canal | 415 | 1.72 | 19 | Friendly | 595 | 1.53 |
| 20 | Comfortable | 335 | 1.39 | 20 | Enjoy | 536 | 1.38 |
| 21 | Free | 318 | 1.32 | 21 | Excellent | 519 | 1.34 |
| 22 | Worth | 290 | 1.20 | 22 | View | 492 | 1.27 |
| 23 | Spacious | 290 | 1.20 | 23 | Awesome | 489 | 1.26 |
| 24 | Staff | 280 | 1.16 | 24 | Spacious | 465 | 1.20 |
| 25 | Shuttle | 278 | 1.15 | 25 | Fun | 444 | 1.14 |
| 26 | Expensive | 256 | 1.06 | 26 | Wonderful | 440 | 1.13 |
| 27 | Awesome | 251 | 1.04 | 27 | Recommend | 435 | 1.12 |
| 28 | Grand | 249 | 1.03 | 28 | Huge | 433 | 1.11 |
| 29 | Clean | 242 | 1.00 | 29 | Luxury | 430 | 1.11 |
| 30 | Sky | 240 | 1.00 | 30 | Canal | 426 | 1.10 |
| 31 | Court | 237 | 0.98 | 31 | Comfortable | 395 | 1.02 |
| 32 | Building | 236 | 0.98 | 32 | Large | 376 | 0.97 |
| 33 | Facilities | 235 | 0.98 | 33 | Helpful | 372 | 0.96 |
| 34 | Wide | 232 | 0.96 | 34 | Bathroom | 336 | 0.87 |
| 35 | Enjoy | 222 | 0.92 | 35 | Parking | 333 | 0.86 |
| 36 | Gambling | 215 | 0.89 | 36 | Free | 326 | 0.84 |
| 37 | Convenient | 213 | 0.88 | 37 | Gorgeous | 324 | 0.83 |
| 38 | Fun | 212 | 0.88 | 38 | Grand | 321 | 0.83 |
| 39 | Crowded | 208 | 0.86 | 39 | Front desk | 320 | 0.82 |
| 40 | Old | 204 | 0.85 | 40 | Pretty | 290 | 0.75 |
| 41 | Price | 202 | 0.84 | 41 | Coffee | 263 | 0.68 |
| 42 | View | 190 | 0.79 | 42 | Atmosphere | 262 | 0.67 |
| 43 | Famous | 187 | 0.78 | 43 | Mall | 250 | 0.64 |
| 44 | Street | 183 | 0.76 | 44 | Fantastic | 245 | 0.63 |
| 45 | Excellent | 182 | 0.76 | 45 | Dining | 244 | 0.63 |
| 46 | Interior | 160 | 0.66 | 46 | Price | 243 | 0.63 |
| 47 | Gorgeous | 160 | 0.66 | 47 | Security | 235 | 0.60 |
| 48 | Egg tart | 159 | 0.66 | 48 | Bed | 233 | 0.60 |
| 49 | Atmosphere | 157 | 0.65 | 49 | Architecture | 225 | 0.58 |
| 50 | Lobby | 155 | 0.64 | 50 | Amenities | 219 | 0.56 |

Note: Green indicates repeated words.

After filtering the collected data, in general, the words related to infrastructure for Macao include "Casino", "Room", "Mall", "Gondola", "Shops", "Canal", "Shuttle", "Restaurants", etc.; meanwhile, those for Las Vegas are "Room", "Casino", "Restaurants", "Shops", "Pool", "Gondola", and so on, reflecting the characteristics of casino hotels. Subsequently, in the high-frequency words, there are many evaluation words, including "Good", "Great", "Amazing", "Best", "Comfortable", etc., indicating that a majority of customer comments are positive. For example, the occurrence frequency of the word "Great" in Macao is 1745 times, accounting for 7.24%; and that in Las Vegas is up to 2913, making up 7.50%. To illustrate this, one of customers in Las Vegas made the following comment: "Awesome Hotel Great Staff Amazing Room and Great Service".

Comparing the high-frequency words of the two casino hotels, they can be divided into 31 repeated words and 19 non-repeated words. In terms of repeated words, among the top ten words, only "Big", existing in Macao, does not overlap in Las Vegas; and only "Service", appearing in Las Vegas, does not show itself in Macao, which shows that they are with a high degree of similarity and that tourists from Las Vegas and Macao have some common and frequent concerns, such as "Casino" and "Room". Moreover, most of their repeated words are in similar rankings, but some words do not follow this rule. To be more specific, "Luxury" comes in at No. 7 in Macao and No. 29 in Las Vegas; "Mall" comes at No. 11 in Macao and No. 43 in Las Vegas; "View" comes at No. 42 in Macao and No. 22

in Las Vegas; and "Staff" ranks 24th and 6th, respectively, in Macao and Las Vegas. These differences indicate that customers in Macao and Las Vegas share common concerns in the above aspects but differ in frequency. Additionally, non-repeated words reveal that customers in Macao pay more attention to price and shopping, while those in Las Vegas concentrate on infrastructure, such as swimming pools.

Consequently, from the view of network visibility and word frequency, customers in Macao pay more attention to hotel "Shopping" and "Price", and those in Las Vegas have a preference for "Physical Environment" and "Service".

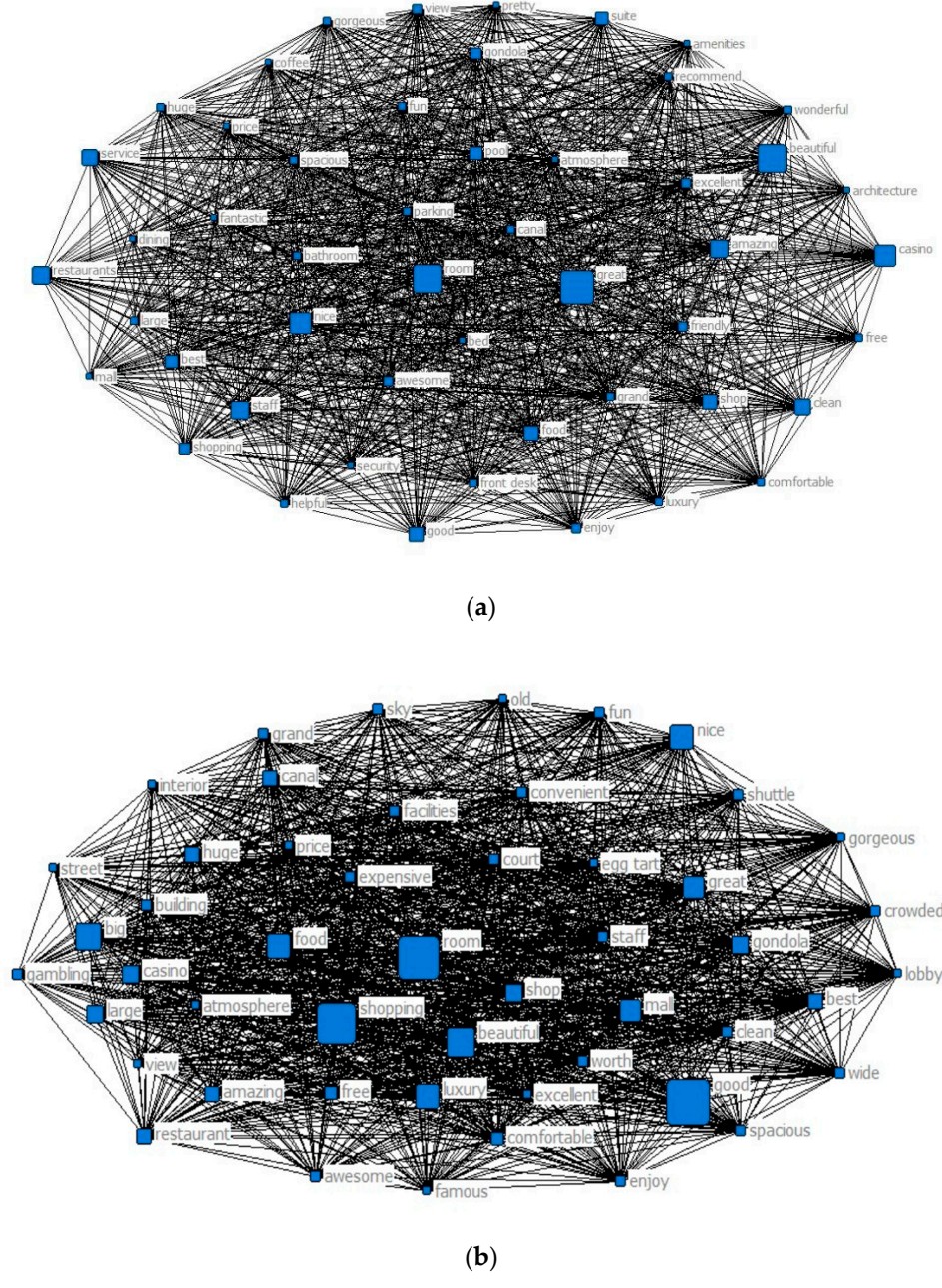

(**a**)

(**b**)

**Figure 2.** Keyword visualization of network analysis. (**a**) The Venetian in Las Vegas; (**b**) The Venetian in Macao.

*4.2. Semantic Network Analysis*

Freeman's degree centrality is an index to measure the connection degree between a node and other nodes in the network; and Eigenvector centrality is a useful indicator to find the most influential central node in the network [81,84]. Tables 2 and 3 make

a comparison of frequency, Freeman's degree centrality, and Eigenvector centrality of high-frequency words for The Venetian Hotel in Las Vegas and Macao, respectively, and compare the two sets of data.

**Table 2.** Comparison of keywords frequency and centrality (The Venetian in Las Vegas).

| | Frequency | | Freeman's Degree Centrality | | Eigenvector Centrality | |
|---|---|---|---|---|---|---|
| | Frequency | Rank | Coefficient | Rank | Coefficient | Rank |
| Room | 4454 | 1 | 28.09 | 1 | 60.75 | 1 |
| Great | 2913 | 2 | 20.32 | 2 | 48.73 | 2 |
| Beautiful | 2395 | 3 | 16.01 | 3 | 39.08 | 3 |
| Nice | 1797 | 4 | 12.62 | 6 | 32.15 | 6 |
| Casino | 1775 | 5 | 14.24 | 4 | 34.27 | 4 |
| Staff | 1438 | 6 | 13.58 | 5 | 33.94 | 5 |
| Amazing | 1418 | 7 | 10.57 | 10 | 26.31 | 10 |
| Service | 1284 | 8 | 10.59 | 9 | 27.85 | 9 |
| Clean | 1239 | 9 | 11.49 | 8 | 30.42 | 7 |
| Restaurants | 1382 | 10 | 11.81 | 7 | 29.56 | 8 |
| Good | 1100 | 11 | 8.85 | 14 | 22.29 | 12 |
| Food | 1056 | 12 | 9.58 | 11 | 24.23 | 11 |
| Shop | 1053 | 13 | 9.18 | 12 | 22.12 | 13 |
| Suite | 942 | 14 | 8.92 | 13 | 21.49 | 14 |
| Best | 877 | 15 | 6.55 | 18 | 16.44 | 18 |
| Pool | 835 | 16 | 7.56 | 15 | 19.05 | 15 |
| Shopping | 712 | 17 | 6.95 | 16 | 16.80 | 17 |
| Gondola | 657 | 18 | 5.18 | 20 | 11.97 | 22 |
| Friendly | 595 | 19 | 6.61 | 17 | 17.31 | 16 |
| Enjoy | 536 | 20 | 4.67 | 23 | 11.22 | 24 |
| Excellent | 519 | 21 | 4.83 | 22 | 12.32 | 20 |
| View | 492 | 22 | 4.19 | 27 | 11.19 | 25 |
| Awesome | 489 | 23 | 3.71 | 32 | 9.55 | 29 |
| Spacious | 465 | 24 | 5.36 | 19 | 14.20 | 19 |
| Fun | 444 | 25 | 3.58 | 34 | 8.95 | 34 |
| Wonderful | 440 | 26 | 3.73 | 31 | 9.39 | 31 |
| Recommend | 435 | 27 | 4.89 | 21 | 12.10 | 21 |
| Huge | 433 | 28 | 4.41 | 24 | 10.95 | 27 |
| Luxury | 430 | 29 | 3.86 | 29 | 9.41 | 30 |
| Canal | 426 | 30 | 4.07 | 28 | 9.19 | 32 |
| Comfortable | 395 | 31 | 4.37 | 26 | 11.12 | 26 |
| Large | 376 | 32 | 3.79 | 30 | 9.77 | 28 |
| Helpful | 372 | 33 | 4.38 | 25 | 11.40 | 23 |
| Bathroom | 336 | 34 | 3.64 | 33 | 9.02 | 33 |
| Parking | 333 | 35 | 3.03 | 37 | 7.13 | 37 |
| Free | 326 | 36 | 3.33 | 36 | 7.67 | 36 |
| Gorgeous | 324 | 37 | 2.83 | 38 | 6.82 | 38 |
| Grand | 321 | 38 | 3.36 | 35 | 7.75 | 35 |
| Front desk | 320 | 39 | 2.60 | 41 | 6.72 | 39 |
| Pretty | 290 | 40 | 2.51 | 42 | 6.46 | 42 |
| Coffee | 263 | 41 | 2.43 | 43 | 6.32 | 43 |
| Atmosphere | 262 | 42 | 2.23 | 47 | 5.72 | 47 |
| Mall | 250 | 43 | 2.31 | 46 | 5.72 | 48 |
| Fantastic | 245 | 44 | 2.40 | 45 | 5.92 | 45 |
| Dining | 244 | 45 | 2.71 | 40 | 6.47 | 41 |
| Price | 243 | 46 | 2.43 | 44 | 6.13 | 44 |
| Security | 235 | 47 | 2.18 | 49 | 5.65 | 49 |
| Bed | 233 | 48 | 2.77 | 39 | 6.72 | 40 |
| Architecture | 225 | 49 | 1.88 | 50 | 4.49 | 50 |
| Amenities | 219 | 50 | 2.19 | 48 | 5.77 | 46 |

**Table 3.** Comparison of keywords frequency and centrality (The Venetian in Macao).

| | Frequency | | Freeman's Degree Centrality | | Eigenvector Centrality | |
|---|---|---|---|---|---|---|
| | Frequency | Rank | Coefficient | Rank | Coefficient | Rank |
| Casino | 2258 | 1 | 24.36 | 1 | 57.73 | 1 |
| Good | 1745 | 2 | 16.03 | 4 | 41.46 | 4 |
| Room | 1637 | 3 | 19.56 | 3 | 46.02 | 3 |
| Shopping | 1548 | 4 | 20.04 | 2 | 51.13 | 2 |
| Beautiful | 1054 | 5 | 9.07 | 9 | 23.49 | 11 |
| Big | 930 | 6 | 10.14 | 7 | 28.59 | 7 |
| Luxury | 900 | 7 | 9.49 | 8 | 25.52 | 8 |
| Nice | 820 | 8 | 7.09 | 15 | 19.63 | 14 |
| Food | 808 | 9 | 13.07 | 5 | 33.03 | 5 |
| Great | 792 | 10 | 8.50 | 11 | 23.90 | 9 |
| Mall | 749 | 11 | 10.45 | 6 | 31.07 | 6 |
| Shop | 595 | 12 | 9.00 | 10 | 23.80 | 10 |
| Large | 560 | 13 | 7.28 | 13 | 21.16 | 13 |
| Gondola | 531 | 14 | 7.19 | 14 | 17.59 | 16 |
| Amazing | 457 | 15 | 4.85 | 21 | 13.30 | 21 |
| Best | 447 | 16 | 4.79 | 22 | 13.74 | 20 |
| Huge | 447 | 17 | 6.53 | 16 | 18.69 | 15 |
| Restaurant | 421 | 18 | 8.27 | 12 | 22.29 | 12 |
| Canal | 415 | 19 | 5.96 | 17 | 14.93 | 18 |
| Comfortable | 335 | 20 | 4.63 | 24 | 12.05 | 24 |
| Free | 318 | 21 | 5.59 | 19 | 14.38 | 19 |
| Worth | 290 | 22 | 3.66 | 27 | 9.58 | 27 |
| Spacious | 290 | 23 | 4.74 | 23 | 12.86 | 22 |
| Staff | 280 | 24 | 4.37 | 25 | 11.50 | 25 |
| Shuttle | 278 | 25 | 4.96 | 20 | 12.27 | 23 |
| Expensive | 256 | 26 | 3.66 | 28 | 9.28 | 28 |
| Awesome | 251 | 27 | 2.10 | 46 | 5.66 | 46 |
| Grand | 249 | 28 | 4.15 | 26 | 10.48 | 26 |
| Clean | 242 | 29 | 3.42 | 31 | 9.16 | 29 |
| Sky | 240 | 30 | 3.63 | 29 | 9.15 | 30 |
| Court | 237 | 31 | 5.64 | 18 | 15.44 | 17 |
| Building | 236 | 32 | 3.19 | 34 | 8.60 | 34 |
| Facilities | 235 | 33 | 3.06 | 35 | 7.89 | 35 |
| Wide | 232 | 34 | 2.09 | 47 | 5.94 | 43 |
| Enjoy | 222 | 35 | 3.39 | 32 | 8.90 | 33 |
| Gambling | 215 | 36 | 2.45 | 40 | 7.14 | 38 |
| Convenient | 213 | 37 | 2.71 | 38 | 7.07 | 39 |
| Fun | 212 | 38 | 2.77 | 37 | 7.54 | 36 |
| Crowded | 208 | 39 | 2.13 | 45 | 5.70 | 45 |
| Old | 204 | 40 | 2.37 | 42 | 6.32 | 41 |
| Price | 202 | 41 | 3.52 | 30 | 8.94 | 32 |
| View | 190 | 42 | 3.34 | 33 | 9.05 | 31 |
| Famous | 187 | 43 | 2.92 | 36 | 7.25 | 37 |
| Street | 183 | 44 | 2.38 | 41 | 6.10 | 42 |
| Excellent | 182 | 45 | 1.63 | 50 | 4.18 | 50 |
| Interior | 160 | 46 | 2.28 | 43 | 5.80 | 44 |
| Gorgeous | 160 | 47 | 1.67 | 49 | 4.22 | 49 |
| Egg tart | 159 | 48 | 1.89 | 48 | 4.58 | 48 |
| Atmosphere | 157 | 49 | 2.17 | 44 | 5.62 | 47 |
| Lobby | 155 | 50 | 2.50 | 39 | 6.58 | 40 |

The results of Las Vegas are shown in Table 2. The top three words, "Room", "Great", and "Beautiful", have the same ranking in frequency, Freeman's degree centrality, and Eigenvector centrality (from first to third). It demonstrates that the frequency, connection, and influence of these three words are very high. Most keywords have similar frequency and influence, but there are some exceptions. Keywords, with relatively high frequency but low degree

centrality and Eigenvector centrality are "View", "Awesome", "Fun", "Wonderful", and "Atmosphere", meaning that they have limited connection with other keywords; meanwhile, those with relatively low frequency but high degree centrality and Eigenvector centrality include "Spacious", "Recommend", "Comfortable", "Helpful", "Dining", and "Bed", implying that they enjoy a high degree of connection with other words.

The results of Macao are shown in Table 3. The frequency, Freeman's degree centrality, and Eigenvector centrality of "Good" and "Room" rank first and third, respectively, depicting that the frequency, connection degree, and influence of these two nodes are very high. Likewise, most keywords in Las Vegas have similar frequency and impact, but several do not. Keywords with relatively high frequency but low degree centrality and Eigenvector centrality include "Beautiful", "Amazing", "Best", "Worth", "Awesome", "Wide", "Crowded", and "Excellent", embodying that they have limited connections to other keywords; meanwhile, those with relatively low frequency but high degree centrality and Eigenvector centrality are "Mall", "Restaurant", "Shuttle", "Court", "Price", "View", "Famous", "Atmosphere", and "Lobby", signifying that they are highly connected to other nodes.

The CONCOR analysis is the connection of the relationship and discovering patterns between words. The greater the similarity of the connection relationship patterns, the higher is the degree of structural equivalence of the other words. It forms clusters that include keywords with similarities to each other [63]. The keywords, which are extracted from the frequency histogram according to the frequency ranking, are utilized to construct a matrix. To visualize the results, NetDraw in the UCINET 6.0 program was applied. The nodes are presented as blue squares, whose sizes represent their frequency, and the network shows the connectivity between them. The result of the CONCOR analysis is shown in Figure 3 with visibility.

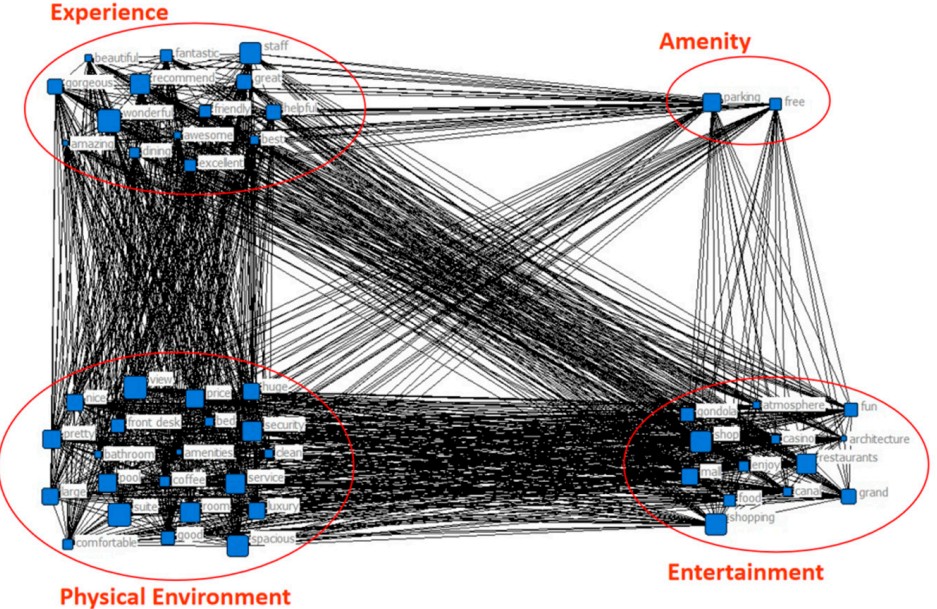

(**a**)

**Figure 3.** *Cont.*

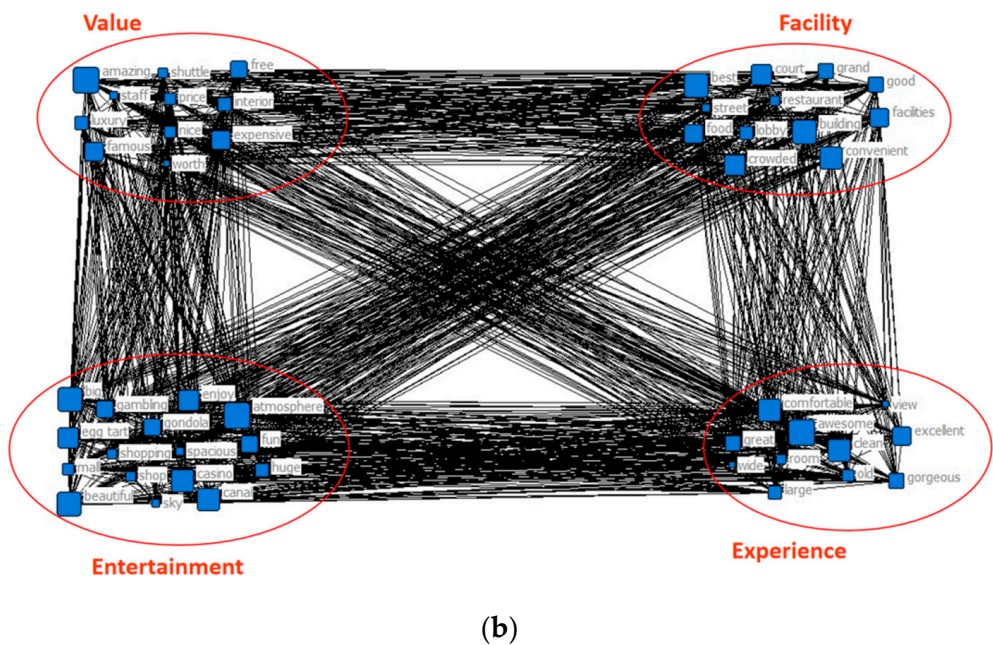

(**b**)

**Figure 3.** Visualization of CONvergence of iterated CORrelation (CONCOR) analysis. (**a**) The Venetian in Las Vegas; (**b**) The Venetian Hotel in Macao.

Las Vegas and Macao each have four groups that are intricately interwoven with each other is shown in Tables 4 and 5. After looking at the words in the group, the group of Las Vegas includes "Experience", "Amenity", "Physical Environment", and "Entertainment", and the group of Macao includes "Value", "Facility", "Entertainment" and "Experience".

**Table 4.** Result of CONCOR analysis (Las Vegas).

| Clusters | Extracted Words | Significant Words |
|---|---|---|
| Experience | beautiful/fantastic/staff/gorgeous/ recommend/great/helpful/friendly/ best/awesome/amazing/dining/ wonderful/excellent | beautiful/fantastic/great/ recommend/helpful/ friendly/awesome/ wonderful/excellent |
| Amenity | parking/free | parking/free |
| Physical Environment | view/nice/huge/bed/pretty/bathroom/ coffee/frontdesk/nice/clean/service/ room/good/luxury/amenities/price/ spacious/comfortable/large/pool/security | view/huge/bed/bathroom/ coffee/front desk/clean/service/room/ amenities/price/spacious/ comfortable/large/pool/security |
| Entertainment | gondola/fun/casino/shop/enjoy/ restaurants/grand/shopping/canal/ architecture/atmosphere/mall/food | gondola/casino/shop/ restaurants/shopping/canal/ atmosphere/mall/food |

**Table 5.** Result of CONCOR analysis (Macao).

| Clusters | Extracted Words | Significant Words |
|---|---|---|
| Value | amazing/shuttle/free/staff/price/ interior/luxury/nice/expensive/ famous/worth | free/price/luxury/ expensive/worth |
| Facility | best/court/grand/street/restaurant/ facilities/building/lobby/good/food/ crowed/convenient | court/street/restaurant/ facilities/building/lobby/ food/convenient |
| Entertainment | big/gambling/enjoy/atmosphere/ gondola/egg tart/ shopping/fun/mall/shop/huge/canal/ casino/beautiful/sky | gambling/gondola/egg tart/shopping/mall/shop/canal/ casino/sky |
| Experience | comfortable/view/awesome/ excellent/clean/wide/great/old/ gorgeous/room/large | comfortable/view/ awesome/excellent/clean/wide/great/ old/gorgeous/large |

For Las Vegas, there are four clusters, namely "Experience", "Amenity", "Physical Environment", and "Entertainment". For Macao, there are also four clusters: "Value", "Facility", "Entertainment", and "Experience". On the one hand, the comparison of the same two groups can yield the following results. In the "Experience", compared with Las Vegas, Macao shows positive comments, such as "Awesome" and "Excellent", yet the word "Old" also appears, implying that the hotel is aging, and that is also a concern for customers. The number of keywords in Las Vegas is more than that in Macao, including not merely adjectives, such as "Wonderful" and "Amazing", but the intention of "Recommend", because positive experience will generate the behavioral intention of recommendation [85]. What is more, it also includes the evaluation of service experience, such as "Staff", "Friendly", "Helpful", and so on. For "Entertainment", both two groups contain "Casino", "Shopping", "Mall", "Gondola", "Canal", and others, all of which not only accord with the characteristics of casino hotel, but conform to the typical characteristics of the Venetian hotel theme and attract the tourists' attention. Nevertheless, in Macao's online review, "Egg tart", a representative food of Macao, has also attracted the attention of guests. On the other hand, the two different groups were analyzed separately. For Las Vegas, the "Physical Environment" includes several physical environment keywords, such as "Bathroom", "Pool", "Bed", "Room", and "Suite"; in addition, it also includes positive comments, such as "Nice", "Pretty", and "Comfortable", proving that customers have positive opinions about the physical environment. Subsequently, the inclusion of "Parking" and "Free" in "Amenity" indicates that they have certain influence on customers. For example, a review states, "Free Parking in Vegas is hard to find and Venetian has that too!", which can be explained by American tourists' preference for self-driving travel; thus, the free parking service will provide convenience for customers. For Macao, "Value" contains "Expensive" and "Luxury", but also "Worth", "Nice", "Shuttle", and "Free", showing that tourists are sensitive to price. Moreover, "Facility" includes "Lobby", "Restaurant", "Facilities", and others, as well as "Convenient", "Best", "Grand", etc. There are no negative keywords, indicating that tourists are satisfied with the infrastructure.

### 4.3. Quantitative Analysis

### 4.3.1. Factor Analysis

The factor analysis is aimed at reducing myriad variables to smaller factors by using the oblique rotation process, so as to find the commonality between these keywords and uncover the relationship of variables through the difference of keywords in hotel reviews. In the factor extraction process, common factor criteria are used, and the minimum factor load is set at 0.400 in the final model. The Eigen value of these factors must be greater than 1.0 and must account for a large proportion of the total variance. In Las Vegas' results, 13 keywords in the four factors covered 46.988% of all variables, while, in Macao, 18 keywords in the five factors covered 46.378% of all variables. For both of them, the hotel experience is generated through the elimination process five times.

Tables 6 and 7 show the results of factor analysis. KMO (Kaiser–Meyer–Olkin) is 0.598 and 0.615, respectively, close to or higher than 0.6. Therefore, the application of exploratory factor analysis is suitable for this study. Bartlett's test values ($X^2$) are 11,743.309 and 21,528.938, respectively. The overall meaning of the correlation matrix is the global significance of the matrix ($p < 0.001$). The two sets of results corroborate that the data do not produce the same matrix and that they are multivariate normal and suitable for factor analysis. The four factors in Las Vegas are named "Physical Environment (Factor 1)", "Entertainment (Factor 2)", "Experience (Factor 3)", and "Amenity (Factor 4)". Factor 1 includes "Bed", "Bathroom", "Room", "Comfortable", and "Suite", all of which are related to physical environment. Factor 2 covers "Shop", "Shopping", and "Restaurants", which are in connection with entertainment. Factor 3 is about experience, including "Staff", "Friendly", and "Helpfully". In addition, Factor 4, concerning amenity, contains "Free" and "Parking". Macao's five factors are named "Entertainment (Factor 1)", "Value (Factor 2)", "Experience (Factor 3)", "Amenity (Factor 4)", and "Physical Environment (Factor 5)".

Factor 1 comprises "Shopping", "Shop", and "Mall', all of which are related to entertainment. Factor 2 consists of "Food", "Court", "Price", and "Expensive", which are related to value. Factor 3 is about experience, including "Room", "Spacious", "Comfortable", and "Clean". Factor 4, concerns amenity, which includes "Shuttle" and "Free". Furthermore, Factor 5 embodies "Canal", "Gondola", "Grand", "Famous", and "Egg tart", which are in connection with physical environment.

**Table 6.** Result of the factor analysis (The Venetian in Las Vegas).

|  | Words | Factor Loading | Eigen Value | Variance (%) |
|---|---|---|---|---|
| Physical Environment | Bed | 0.709 | 2.021 | 13.118 |
|  | Bathroom | 0.594 |  |  |
|  | Room | 0.583 |  |  |
|  | Comfortable | 0.571 |  |  |
|  | Suite | 0.451 |  |  |
| Entertainment | Shop | 0.878 | 1.709 | 12.308 |
|  | Shopping | 0.828 |  |  |
|  | Restaurants | 0.490 |  |  |
| Experience | Staff | 0.759 | 1.483 | 11.75 |
|  | Friendly | 0.746 |  |  |
|  | Helpful | 0.656 |  |  |
| Amenity | Free | 0.823 | 1.365 | 9.813 |
|  | Parking | 0.821 |  |  |
| Total variance (%) = 46.988 | | | | |
| KMO (Kaiser Meyer Olkin) = 0.598 | | | | |
| Bartlett chi-square ($p$) = 11,743.309 ($p < 0.001$) | | | | |

**Table 7.** Result of the factor analysis (The Venetian Hotel in Macao).

|  | Words | Factor Loading | Eigen Value | Variance(%) |
|---|---|---|---|---|
| Entertainment | Shopping | 0.927 | 2.605 | 14.470 |
|  | Shop | 0.905 |  |  |
|  | Mall | 0.556 |  |  |
| Value | Food | 0.815 | 1.604 | 8.912 |
|  | Court | 0.791 |  |  |
|  | Expensive | 0.432 |  |  |
|  | Price | 0.423 |  |  |
| Experience | Room | 0.697 | 1.453 | 8.072 |
|  | Spacious | 0.659 |  |  |
|  | Comfortable | 0.576 |  |  |
|  | Clean | 0.503 |  |  |
| Amenity | Shuttle | 0.862 | 1.392 | 7.731 |
|  | Free | 0.858 |  |  |
| Physical Environment | Canal | 0.684 | 1.295 | 7.193 |
|  | Gondola | 0.614 |  |  |
|  | Grand | 0.495 |  |  |
|  | Famous | 0.460 |  |  |
|  | Egg tart | 0.413 |  |  |
| Total variance (%) = 46.378 | | | | |
| KMO (Kaiser–Meyer–Olkin) = 0.615 | | | | |
| Bartlett chi-square ($p$) = 21528.938 ($p < 0.001$) | | | | |

When comparing the two groups of data, there are four same factors, namely "Entertainment", "Experience", "Amenity", and "Physical Environment". "Entertainment" includes both "Shop" and "Shopping", but Las Vegas has "Restaurants", while Macao has "Mall". As for "Experience", "Staff", "Friendly", and "Helpful" exist in Las Vegas,

conveying that customers pay more attention to the service experience of employees; meanwhile, "Room", "Spacious", "Comfortable", and "Clean" appear in Macao, indicating that their customers lay emphasis on the size, cleanliness and comfort of rooms. Furthermore, the comparison of their "Amenity" uncovers that customers in Las Vegas pay more attention to the free parking service in hotels, while those in Macao prefer the free shuttle bus provided by hotels. Moreover, Macao also possesses the factor of "Value". There are comments such as "Price" and "Expensive" on "Food" and "Court" in Macao, connoting that tourists in Macao attach importance to "Value" in addition to the common concern for "Entertainment", "Experience", and "Amenity".

### 4.3.2. Linear Regression Analysis

After factor analysis, customer experience and satisfaction in Las Vegas and Macao were analyzed through regression analysis, as shown in Tables 8 and 9. In Table 8, there are four independent variables in Las Vegas, namely "Physical Environment" (PE), "Entertainment" (En), "Expensive" (Ex), and "Amenity" (A), and one dependent variable, namely customer satisfaction (CS). The total variance, explained by the four predictors of customer satisfaction (CS), is 8.9% ($R^2 = 0.089$) and the standard error of the estimated value is 0.866; the correlation between independent variables and the dependent variable is quite low. In Table 7, Macao has five independent variables, namely "Physical Environment" (PE), "Entertainment" (En), "Value" (V), "Expensive" (Ex), and "Amenity" (A), and one dependent variable, namely customer satisfaction (CS). The total variance explained by the five predictors is 11.8% ($R^2 = 0.118$), and the standard error of the estimated value is 0.907. The correlation between independent and dependent variables is quite low, because many factors that affect customer experience and satisfaction may not be included in corresponding factors due to their low frequency in online hotel reviews. In regression models, it is impossible to estimate output variables by covering all relevant variables, such as opinions in text mining data; thus, the $R^2$ value may be very low. For instance, Kim and Noh used regression analysis and factor analysis to study online reviews of washing machines, and the $R^2$ value was 12.5% [86].

**Table 8.** Results of linear regression analysis (The Venetian in Las Vegas).

| Model | Unstandardized Coefficient | | Standardized Coefficient | t |
|---|---|---|---|---|
| | **B** | **SE** | **Beta** | |
| (Constant) | 4.573 | 0.009 | | 489.324 |
| Physical Environment (PE) | −0.137 | 0.009 | −0.151 | −14.668 *** |
| Entertainment (En) | 0.189 | 0.009 | 0.208 | 20.239 *** |
| Experience (Ex) | 0.055 | 0.009 | 0.060 | 5.865 *** |
| Amenity (A) | 0.127 | 0.009 | 0.140 | 13.598 *** |

Notes: Dependent variable, customer satisfaction (CS); $R^2 = 0.090$; adjusted $R^2 = 0.089$; F = 211.024; *** $p < 0.001$.

**Table 9.** Results of linear regression analysis (The Venetian in Macao).

| Model | Unstandardized Coefficient | | Standardized Coefficient | t |
|---|---|---|---|---|
| | **B** | **SE** | **Beta** | |
| (Constant) | 4.494 | 0.009 | | 486.264 |
| Entertainment (En) | 0.009 | 0.009 | 0.011 | 0.953 |
| Value (V) | −0.051 | 0.009 | −0.061 | −5.487 *** |
| Experience (Ex) | −0.017 | 0.009 | −0.020 | −1.809 |
| Amenity (A) | −0.007 | 0.009 | −0.009 | −0.786 |
| Physical Environment (PE) | 0.25 | 0.009 | 0.030 | 2.687 * |

Notes: Dependent variable, customer satisfaction (CS); $R^2 = 0.005$; adjusted $R^2 = 0.005$; F = 8.424; * $p < 0.05$, *** $p < 0.001$.

In the Las Vegas' results, "Entertainment" (En, $\beta$ = 0.208, $p$ < 0.001), "Expensive" (Ex, $\beta$ = 0.060, $p$ < 0.001), and "Amenity" (A, $\beta$ = 0.140, $p$ < 0.001) are positively correlated with customer satisfaction. However, "Physical Environment" (PE, $\beta$ = −0.151, $p$ < 0.001) is negatively correlated with customer satisfaction, meaning that the performance of "Physical Environment" harmed customer experience, and customer satisfaction was reduced. For example, one review stated the following: "Not a good experience. My room was a non-smoking room but had clearly been smoked in previously. It was filled with smoke and no apparent effort to remove it prior to my arrival". In order to estimate the possible correlations among the predictors, a multicollinearity statistic was conducted. The tolerance level is less than 1.00, and the variance inflation factor (VIF) of the predictors is between 1.00 and 1.10, respectively; that is, the predictors are not significantly correlated to each other. Therefore, based on unstandardized $\beta$, the regression equation can be expressed as follows:

$$CS = 4.573 − 0.137PE^* + 0.189En^* + 0.055Ex^* + 0.127A^*$$

"Entertainment" (En) factor holds the highest standardized coefficients, meaning the entertainment of the hotel is the most important factor associated with customer satisfaction significantly. Reviews such as "The shear number of amazing restaurants alone is a foodies dream" and "The shop were fun to visit overall a very good time for the family" are related to the hotel experience based upon "Entertainment" attributes.

For Macao, "Physical Environment" (PE, $\beta$ = 0.030, $p$ < 0.05) positively correlates with customer satisfaction, while "Value" (V, $\beta$ = −0.061, $p$ < 0.001) is negatively correlated with customer satisfaction, meaning that the performance of "Value" harmed customer experience, and customer satisfaction was reduced. For example, one review stated the following: "The food is quite expensive but not very good". In order to estimate the possible correlations among the predictors, a multicollinearity statistic was conducted. The tolerance level is less than 1.00, and the variance inflation factor (VIF) of the predictors is between 1.00 and 1.10, respectively; that is, the predictors do not have a significant correlation to each other. Therefore, based on unstandardized $\beta$, the regression equation can be expressed as follows:

$$CS = 4.494 + 0.009En − 0.051V^* − 0.17Ex − 0.007A + 0.25PE^*$$

The "Physical Environment" (PE) factor holds the highest standardized coefficients, indicating that the amenity of the hotel is the most important factor associated with customer satisfaction significantly. Reviews such as "Amazing third floor where you can enjoy the canals in a gondola" and "Don't forget the famous egg tart" are in relation with the hotel experience based upon "Physical Environment" attributes.

### 4.4. Summary of Results

In order to investigate the eWOM of casino hotels, this study used visualization and data analysis to find out the key points that customers value and the factors that affect customer satisfaction; moreover, it analyzed the eWOM of Las Vegas and Macao casino hotels, so as to improve casino-hotel customers' online review experience and satisfaction. For online hotel review analysis, the first step is to extract keywords through text mining, and the second step is to calculate the frequency of customers, using words. The study conducted an analysis of the frequency, correlation degree, and Eigenvector centrality of the top 50 frequent words for the two cases respectively; meanwhile, it detected their connection degree and the most affected keywords. Through CONCOR analysis, the two sets of keywords were divided into four groups, respectively. For Las Vegas, the keywords were grouped into "Experience", "Amenity", "Physical Environment", and "Entertainment", while, for Macao, they were divided into "Value", "Facility", "Entertainment", and "Experience", all of which were visualized by using NetDraw to map networks and nodes in UCINET6.0. In addition, factor analysis and linear regression analysis were carried out to extract the factors affecting customer satisfaction and to understand the relationship

between them. For one thing, factor analysis reduces the dimensions of the original 50 variables. It extracted 13 keywords from Las Vegas that were divided into four factors, namely "Physical Environment", "Entertainment", "Experience", and "Amenity"; 18 keywords were extracted from Macao that were classified into five factors, namely "Entertainment", "Value", "Experience", "Amenity", and "Physical Environment". For another, there is a correlation between CONCOR analysis and factor analysis. Linear regression analysis takes customer score as the basis of customer satisfaction and obtains the relationship between impact factors and satisfaction.

## 5. Discussion

On the one hand, this study can find out the similarities of customers from different cultural backgrounds in their casino-hotel experience. In the frequency analysis, among the top 10 keywords of Las Vegas and Macao, there were four identical words, namely "casino", "room", "beautiful", and "great", showing that, for casino hotels, casinos and rooms are the main concerns of tourists, who also have a positive impression of "beautiful" and "great" on such hotels. Moreover, there are 28 identical keywords in the two groups, accounting for 56%. In the comparison of the ranking of repeated words, most of them are relatively close, reflecting that customers in different regions have common concerns about their casino-hotel experience, but they differ in frequency. Additionally, in CONCOR analysis, in the same groups, "Entertainment" contains the same keywords: "casino", "mall", "shop", "shopping", "gondola", "canal", "atmosphere", etc.; while "Experience" includes "great", "excellent", "gorgeous", etc., indicating that tourists from different regions have the same demand for casino and shopping, as well as themed features, and they have some common positive comments on the casino hotel experience. According to the factor analysis results, the four factors of "Physical Environment", "Entertainment", "Experience", and "Amenity" are the same, and only one factor is different. However, there is a big difference between the two groups in the results of the linear regression.

On the other hand, it can compare the differences of customers from different cultural backgrounds in their casino-hotel experience. In the frequency analysis, among their top 10 words, different keywords of Las Vegas are "Staff", "Service", "Amazing", "Clean", "Restaurants", etc., appear; meanwhile, Macao contains 22 different keywords, such as "Shopping", "Big", "Luxury", "Food", and "Good", indicating that Chinese and American tourists have different priorities in casino hotel experience. Moreover, in the CONCER analysis, the different groups are as follows: Las Vegas contains "Amenity" and "Physical Environment"; and Macao includes "Facility" and "Value", implying that Chinese and American customers' experience in casino hotels is different in these four dimensions. Furthermore, according to the results of the factor analysis, there is one more factor, "Value", in Macao than in Las Vegas, which reveals that customers of Macao's casino hotels pay more attention to value. However, it is worth noting that, in the results of the linear regression analysis, the results of their influence on customer satisfaction in Chinese and American casino hotels are completely different. For Las Vegas, "Entertainment", "Experience", and "Amenity" have a positive correlation with customer satisfaction, while "Physical Environment" has a negative correlation with customer satisfaction; among them, "Entertainment" is the most influential factor. For Macao, only "Physical Environment" and "Value" have an impact on customer satisfaction; that is, "Physical Environment" has a positive correlation, while "Value" has a negative correlation, and "Physical Environment" is the most influential factor.

## 6. Conclusions

First and foremost, customers from different cultural backgrounds still share similarities in casino-hotel experience. In casino hotels, gaming attracts the most attention of customers, and gaming revenue is also the main part of hotels' revenue. Thus, in addition to maintaining rooms, restaurants, swimming pools, and other infrastructure, casino hotels still need to develop manifold entertainment projects, so as to attract gambling customers.

Younger cohorts have an increased preference for social-based gambling activities and greater preference for non-gaming offers [28]. The Venetian Hotel, for example, features Venetian-style architecture and immersive experience, covering gambling, accommodation, catering, shopping, exhibition, and other functions, offering customers an all-round diverse experience and leaving them a representative impression. This model is being employed by more and more casino hotels, such as The Parisian hotel and Wynn Palace hotel.

Subsequently, for customers of Las Vegas casino hotels, their attention is relatively focused on the hotel's infrastructure, convenient service, and hotel staff services. For instance, driving is one of the major modes of transportation for visitors to Las Vegas, but customers mentioned in reviews that there are very few free-parking service in Las Vegas. Since the case hotel has free parking service, customers have left a deep impression and improved their satisfaction. Customers' experience will affect their behavioral intention, which is mainly reflected in customers revisiting and giving recommendations [84]. It suggests that casino hotels should strengthen their awareness of serving customers, discover customers' needs, and formulate corresponding service measures, which may achieve more than expected results. At the same time, the maintenance of hotel infrastructure and other physical environment is also very important, and the lack of infrastructure maintenance will directly have a negative impact on customer satisfaction. For Macao casino hotels, managers should grasp the marketing of products. To be more specific, they can make use of customers' sensitivity to price and create marketing measures to enhance customer stickiness, such as membership cards based on point and holiday discount promotion activities, to stimulate their consumption desire.

In practice, online review analysis can be used as a marketing tool for managers, as customer reviews can directly influence hotel revenue, and they are an important basis for service improvement and promotion. Moreover, the analysis provides the importance level of these service attributes; thus, the hotel industry can allocate resources accordingly. What is more, online review analysis can provide a reliable evaluation of satisfaction. Likewise, those in the hotel industry can use this method to analyze their competitors' customer reviews so that they can make sustainable strategic marketing decisions against competitors.

However, the study shows the limitations of data collection. First of all, the data collected in this study are limited, because they only focus on The Venetian Hotel in Las Vegas and Macao. Secondly, the collected text was analyzed according to the frequency of individual words; thus, it is difficult to understand the added meaning of words. In future studies, researchers should collect more casino hotels' online reviews to generalize the survey results; moreover, further positive and negative analyses and sentiment analyses are expected to better understand customer experience and satisfaction. In addition, the specific impact of the casino on customer satisfaction has not been measured in the results of this study. In view of the importance of the casino to the operation of casino hotels, future research in this area is needed, as it can provide a stronger strategy for the casino hotel industry.

**Author Contributions:** H.-S.K. and M.T. constructed the research; M.T. analyzed online review data with the instruction of H.-S.K.; M.T. wrote the draft, and English editing was conducted by H.-S.K. All authors have read and agreed to the published version of the manuscript.

**Funding:** This research received no external funding.

**Institutional Review Board Statement:** Not applicable.

**Informed Consent Statement:** Not applicable.

**Data Availability Statement:** Not applicable.

**Conflicts of Interest:** The authors declare no conflict of interest.

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
