# Peer review of "An Exploratory Study of Electronic Word-of-Mouth Focused on Casino Hotels in Las Vegas and Macao"

_information, doi:10.3390/info13030135_

Round 1

Reviewer 1 Report

Figure 2. Research Process must be corrected.

  1. Follow the rules for visual presentation of the process.
    SCTM 3.0 is not the result of an online review, but a tool used to conduct an online review.
  2. The same mistakes are repeated in others.
  3. Linear Regression Analysis from the results of Semantic Network Analysis and factor analysis.
    As now drawn it looks completely independent.
    It is necessary to draw it one level lowe

Author Response

  • We do appreciate your instrumental and meaningful comments on the improvement of this manuscript. Based on your suggestions, we have revised the structural relationships of Figure 2. It is indeed to be clearly shown.

Reviewer 2 Report

The intro section was well presented including the overall importance and the purpose of the research. The topic of the research is interesting and would be meaningful in the hospitality and tourism literature.

The lit review demonstrated a proper review of the previous studies on the subjects.

The method section described the step-by-step process of the analysis. But, in the intro section, the data collection period was Oct 2017 - Oct 2021, but here in this section Oct 2017 - Sep 2021. The authors need to check which one is correct.

The results section is well presented with tables, figures, and texts. But, one concern is the negative b coefficients in the regression analysis. The authors briefly mentioned the results in the results and discussion sections, but how would the authors specifically interpret the "negative" coefficients (especially significant ones)? For example, In Las Vegas, PE has a significantly negative impact on CS. What does it mean? 

Overall, the manuscript is well organized and presented. 

Line 570: CONCOR?

Author Response

Thank you for your comment on our paper. We do appreciate your instrumental and meaningful comments on the improvement of this manuscript. Thank you so much.

1. The collection period that has to be checked was Oct 2017 - Sep 2021 and we have corrected in the manuscript at line 63.

2. Based on your suggestions, we have added the interpretation of negatively correlated and combined with the content of customer online review to reflect in each case hotel.

3. We are truly sorry if there are any grammar or spelling mistakes. Thank you for your kind reminder, the manuscript has been proofread by the authors.